# Real-time UVMS torque distribution algorithm based on weighting matrix

**Yecheol Moon[1], Jongin Hong[1], Sangrok Jin[2], Jangho Bae[3]\*, TaeWon Seo[1]\***

**1** School of Mechanical Engineering, Hanyang University, Seoul, South Korea, **2** School of Mechanical Engineering, Pusan National University, Busan, South Korea, **3** GRASP Lab, University of Pennsylvania, Philadelphia, PA, United States of America

\* taewonseo@hanyang.ac.kr (TS); jangho.bae91@gmail.com (JB)

## Abstract

This study presents a real-time algorithm for even distributing the torque burden on the parallel manipulator with an autonomous underwater vehicle (AUV) through the cooperation of the AUV and manipulator. For the redundant resolution of the underwater vehicle manipulator system (UVMS), we used the weighting matrix of the weighted pseudo inverse for kinematic and dynamic modeling. We made dynamic and kinematic modeling using the force distribution characteristics of parallel manipulators. Using the parallel manipulator's model, the weighting matrix was changed every second to share the manipulator torque with the AUV. The Taguchi method was used to reduce the calculation time for real-time calculation and to perform valve rotation operations with as little torque as possible even in an underwater environment where it is difficult to determine any cause of errors. To demonstrate the effectiveness of this algorithm, we experimented with valve rotation in water using the UVMS. Analysis of the experimental results revealed that the manipulator torque load was greatly reduced due to the AUV load distribution.

## 1. Introduction

Various underwater robots have been developed to control the underwater situations inaccessible to humans. An autonomous underwater vehicle (AUV) is an underwater robot that performs autonomous control without any external input from an operator. It has been in the development stage since the 1950s, as reported by Bogue [1]. As summarized in Yuh's research, related technologies such as data fusion, fault tolerance, and obstacle avoidance systems have been developed to control the AUVs [2]. Sivčev et al. reported that various underwater manipulators have been developed to control underwater situations [3].

The system using the underwater manipulator attached to the AUV is called the underwater vehicle manipulator system (UVMS). As reported by Antonelli in his publication, UVMS modeling, and basic research have already been performed [4], based on which, a new study of manipulators was conducted. UVMS has been studied in various fields based on modeling and the impact of UVMS [4]. Casalino et al. studied the issues with manipulator adjustment and the non-holonomic AUV, where the entire system is identified as UVMS manipulation tasks [5]. Lapierre et al. introduced a control method that corrects the position error in the platform

**Funding:** This work was supported by the Basic Science Research Program through the National Research Foundation of Korea(NRF) funded by the

Ministry of Science and ICT(NRF-
2017R1A2B4002123), the International
Collaboration Research Program through the NRF
funded by the Ministry of Science and ICT(NRF-
2017K1A3A1A19071037), and partly by the
Human Resources Program in Energy Technology
of the Korea Institute of Energy Technology
Evaluation and Planning(KETEP) granted financial
resource from the Ministry of Trade, Industry &
Energy, Republic of Korea (No. 20204030200100).

**Competing interests:** The authors have declared
that no competing interests exist.

with the force control loop included in the position control loop [6]. Farivarnejad et al. studied a control system for the operation of h-task-priority inverse kinematics approach to redundancy resolution heavy cylinders, which must be fixed to underwater structures with dual-arm UVMS [7]. Sagara et al. proposed a solved acceleration control method for UVMS based on a workspace consisting of an AUV and end-effector position and orientation [8]. Youakim et al. examined a general approach to motion planning in underwater UVMS and presented a benchmark for comparison of algorithms [9]. Han et al. suggest a performance index designed to minimize the restoring moments of the UVMS during manipulation for redundancy resolution [10]. Casavola et al. present a fault-tolerant adaptive control allocation scheme for over actuated systems subject to loss of effectiveness actuator faults [11].

Cooperation of the AUV and the manipulator is effective in reducing the load on the manipulator. The manipulator joint load is reduced by distributing the force using the thruster power of the AUV, which can improve the overall performance. Han et al. controlled the UVMS manipulator and AUV completely to minimize the moment of restoration of the entire system [12]. According to Inoue et al., to withstand the external forces, a full-body control method was applied to a mobile manipulator [13]. However, the method of distributing the workload between joints of manipulators and the thrusters of the AUV has not been studied.

In order to present an efficient collaboration method of UVMS, we have conducted basic studies of each AUV and manipulator. In a previous study, Bae et al. studied the cooperation between an underwater manipulator and an AUV for a valve turning operation [14, 15]. The UVMS consists of two parallel manipulators with three joints and an underwater robot with tilting thrusters for redundant actuation (AURORA) with four thrusters [16]. The overall appearance of UVMS is shown in Fig 1. Dynamic modeling of the UVMS was constructed by combining the methods studied by Spong [17] and Schjølberg [18]. The weighted pseudo inverse was used to control the redundant manipulator. In the weighted pseudoinverse, a weighting matrix is used through which the driving element and driving force can be selected. We experimented with a real water tank to evaluate the performance of the valve control in this system.

Based on Bae's study on UVMS, we started this study to improve the load distribution method that could further lower the overall torque. Errors that can occur during an underwater operation are the geometric errors owing to free-floating, valve friction because of rust, thruster modeling error, joint backlash, and disturbance, as discussed in previous studies.

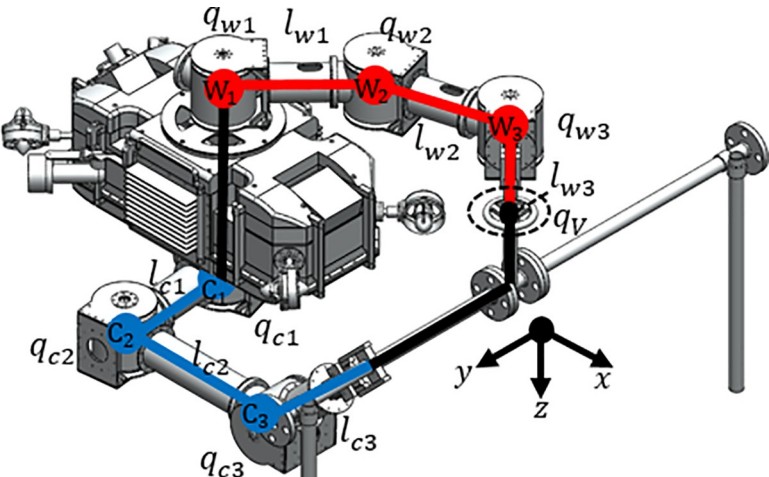

**Fig 1. Overall form and kinematic diagram of UVMS.**

Even if there is such an error, an algorithm that can effectively distribute the torque is required. Besides, by selecting an appropriate weighting matrix used in weighted pseudoinverse, torque burden can be reduced to the maximum extent. Finding a weighting matrix, which can reduce this torque burden to the maximum extent, will enable an effective torque distribution. In summary, even if there is an error, we need an algorithm that can achieve sufficient distribution and determine the optimal weighting matrix.

There are many geometric and non-geometric compensation methods for manipulator errors; however, it is difficult to apply these techniques to underwater manipulators. To correct the geometric error in the manipulator, the error is corrected through modeling based on the Denavit–Hartenberg (DH) model [19], S Model [20], complete and parametrically continuous (CPC) model [21], Zero reference [22], and product of exponentials (POE) [23]. Non-kinematic errors are too diverse and difficult to solve [24, 25]. Non-kinematic errors have been studied in specific fields such as the deformation of gears [26], links, and joints [27]. In addition, laser trackers [28], 3D measuring devices [29], and cameras [30] that measure the errors in these methods are difficult to use underwater because of their large external measuring devices. Similarly, in fluid mechanics, problems that are difficult to solve are often modeled through analysis because there are many factors to consider [31, 32]. However, in large-scale multi-system UVMS, it is difficult to apply due to the limitations of the analysis device. To summarize, it is difficult to correct and measure the cause of torque difference in an underwater manipulator. Therefore, the joint torque should be lowered without modeling.

To determine the weighting matrix without a model and reduce calculating time, the design of the experiment (DOE) and the Taguchi method were used together. DOE is a method for studying the factors affecting the experiments [33]. To reduce the number of experiments, it includes the concept of reducing the number of experiments through orthogonal arrays, the concept of design variables and objective functions, and sensitivity analysis. The Taguchi method is a robust optimal design method that maximizes product performance under any user condition [34]. The key to Taguchi's method is to ensure performance even when users use the product in unexpected user conditions. Therefore, the Taguchi method can be usefully used for control that is difficult to predict due to various causes of errors. Among the Taguchi methods, the algorithm was constructed by mixing it with the signal-to-noise (S/N) ratio. Similarly, there are examples of using DOE and Taguchi methods to reduce the analysis time. Lim et al. presented the procedure and results of multi-purpose optimization of a 7 degrees of freedom manipulator for performance related to global conditioning, operability, and structural length [35]. Nikdel et al. demonstrated that the Taguchi method could be combined with various methods through the hybrid Taguchi-genetic algorithm to improve the joint robot control efficiency [36]. To summarize, DOE and Taguchi methods were used in manipulators in combination with other methods and the combination was used to improve the performance and reduce the computation time.

In this study, a real-time algorithm that divides the manipulator's torque load with an AUV through the cooperation of the AUV and the manipulator was proposed and tested. To reduce the computation time, we used the DOE and the S/N ratios. By changing the weighting matrix using the manipulator joint angle and model every second, a weighting matrix element with the highest S/N ratio was determined. Even if the underwater situation changes rapidly and is unknown, the weighting matrix is calculated based on the Taguchi method at every moment. This weighting matrix is selected in the best-case scenario, and then the torque calculation and manipulator control are performed. An experiment was conducted to prove the effectiveness of this algorithm by comparing it with the performance of the valve operation used in the previous Bae's study. This experiment was conducted in an underwater environment of a large tank filled with water by attaching an actual commercial valve and pipe to the tank. In the

experiment, the valve was rotated 90˚, similar to Bae's study, and the torque was compared with the case in the previous study where the weighting matrix was fixed. Even if the specific error factor is unknown, DOE and the level average analysis of DOE are executed in real-time, so that the torque can be distributed efficiently. To organize the results of the experiment, we refer to papers that analyze the results by changing variables such as speed and angle using commercially available products [37–39].

The contribution of this study is to present a method of finding an appropriate weighting matrix in real-time for the weighting matrix of the conventional pseudoinverse-based UVMS collaborative control. In this paper, we describe the cooperative UVMS in the previous study based on the traditional pseudo-inverse weighting matrix used. Based on the same previous UVMS, we introduce an algorithm that can control the UVMS by selecting an appropriate weight matrix in real-time that can lower the manipulator torque even when the underwater environment changes rapidly. This study consists of the previous UVMS, main algorithms, experiments, and conclusions. Section 2 describes the kinematic and dynamic modeling of UVMS used in previous studies. Section 3 introduces a real-time algorithm that determines a weighting matrix between the two subsystems: the AUV and the manipulator. Section 4 shows the results of the experiments compared to the environment in the previous study. Finally, the conclusion is presented in Section 5.

## 2. Underwater manipulator analysis and control

### 2.1. Kinematics and dynamics of UVMS

The UVMS is composed of underwater manipulators attached to the top and bottom of the AURORA. AURORA moves and shares the force for underwater operations using the four rotatable thrusters attached to the main body. The shape of AURORA is depicted in Fig 2, the overall appearance of the manipulator is depicted in Fig 3.

The UVMS is designed to turn the underwater handle valve. UVMS is used to clamp a fixed structure during valve rotation. Each manipulator has three joints, and all three joints possess built-in motors and torque sensors. Since the torque sensor is directly attached to the motor, the torque generated by the motor is measured immediately. Since the torque sensor can

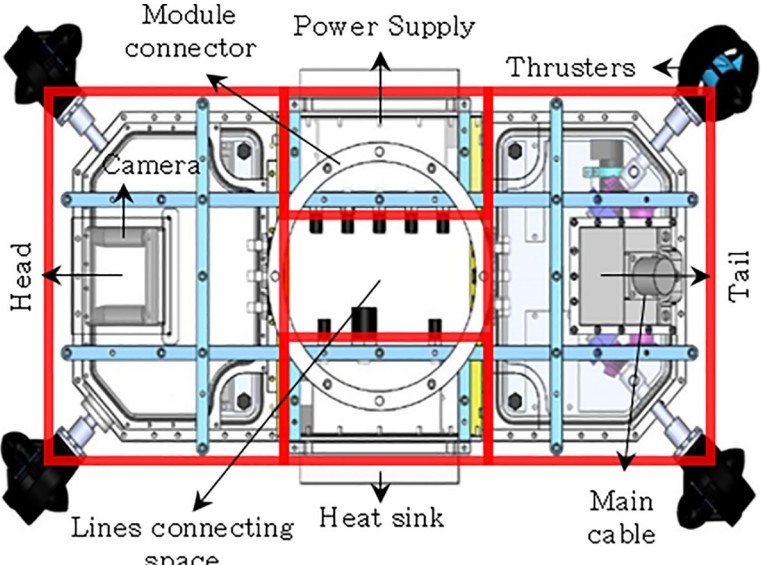

**Fig 2. Structure diagram of an underwater robot with tilting thrusters for redundant actuation (AURORA).**

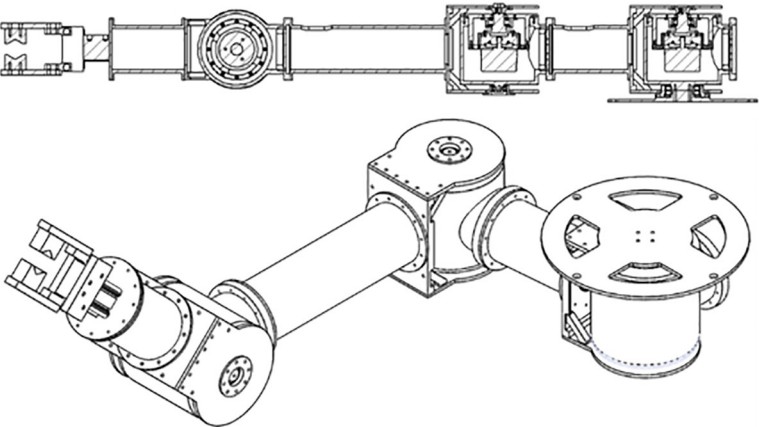

**Fig 3. Assembly appearance of a manipulator.**

measure up to 50Nm with a resolution of 0.05Nm, it is suitable for measuring the joint torque of a robot in this study. The gripper used in these manipulators has a gripping moment of 12.4Nm and a gripping force of up to 180N. It is within the allowable range of force and torque that can occur during valve operation and it is possible to reduce the possibility of errors due to slip of the gripper. Even in actual operation, no-slip occurred in the gripper. The manipulator that grabs and turns the valve is called the working manipulator. Another manipulator that clamps a fixed pipe is called the clamping manipulator. This manipulator is a parallel manipulator because the manipulator forms a closed curve. Each joint is W1, W2, W3, C1, C2 and C3 and shown in Fig 1. $\boldsymbol{q}_w$ ($q_{w1}$, $q_{w2}$, $q_{w3}$) is the angle of each working manipulator. $\boldsymbol{q}_c$($q_{c1}$, $q_{c2}$, $q_{c3}$) is the angle of the clamping manipulator. The lengths of the links connecting each joint are expressed as $l_{w1}, l_{w2}, l_{w3}, l_{c1}, l_{c2}$, and $l_{c3}$. $l_{wn}$ means length between joints $W_n$ and $W_{n+1}$. $l_{cn}$ means length between joints $C_n$ and $C_{n+1}$. The position and orientation of the body can be calculated by assuming a virtual joint with 6 degrees-of-freedom based on the ground, which is expressed as $\boldsymbol{\eta} = [x\ y\ z\ \phi\ \theta\ \psi]^T$. The angle of the valve rotated is denoted by $q_V$. The features of these joints are summarized in Table 1.

The active joint $\boldsymbol{q}_r$ is a joint controlled by a motor. The independent joint $\boldsymbol{q}_u$ is the joint that determines the configuration of the system. Joints denoted as $\boldsymbol{q}_{all}$ represent all the joints of parallel manipulators. We describe the kinematics between the total joint and independent joint using as

$$\dot{\boldsymbol{q}}_r = \boldsymbol{\Gamma}\dot{\boldsymbol{q}}_u \tag{1}$$

$$\dot{\boldsymbol{q}}_{all} = \boldsymbol{\Lambda}\dot{\boldsymbol{q}}_u. \tag{2}$$

$\boldsymbol{\Lambda}$ and $\boldsymbol{\Gamma}$ are the constraint Jacobian to convert $\dot{\boldsymbol{q}}_u$ to $\dot{\boldsymbol{q}}_r$ and $\dot{\boldsymbol{q}}_{all}$. The approximate trajectory of the UVMS is depicted in Fig 4. As the kinematic studies were covered in previous researches, they were omitted from this study.

**Table 1. S/N major categories and vectors of joints used in manipulators.**

| Joint category | Joint vectors |
|---|---|
| Independent Joints $\boldsymbol{q}_u$ | $\boldsymbol{q}_{w1}$, $\boldsymbol{q}_{w2}$, $\boldsymbol{q}_{w3}$ |
| Active Joints $\boldsymbol{q}_r$ | $\boldsymbol{\eta}$, $\boldsymbol{q}_w$, $\boldsymbol{q}_c$ |
| All Joints $\boldsymbol{q}_{all}$ | $\boldsymbol{\eta}$, $\boldsymbol{q}_w$, $\boldsymbol{q}_c$, $\boldsymbol{q}_V$ |

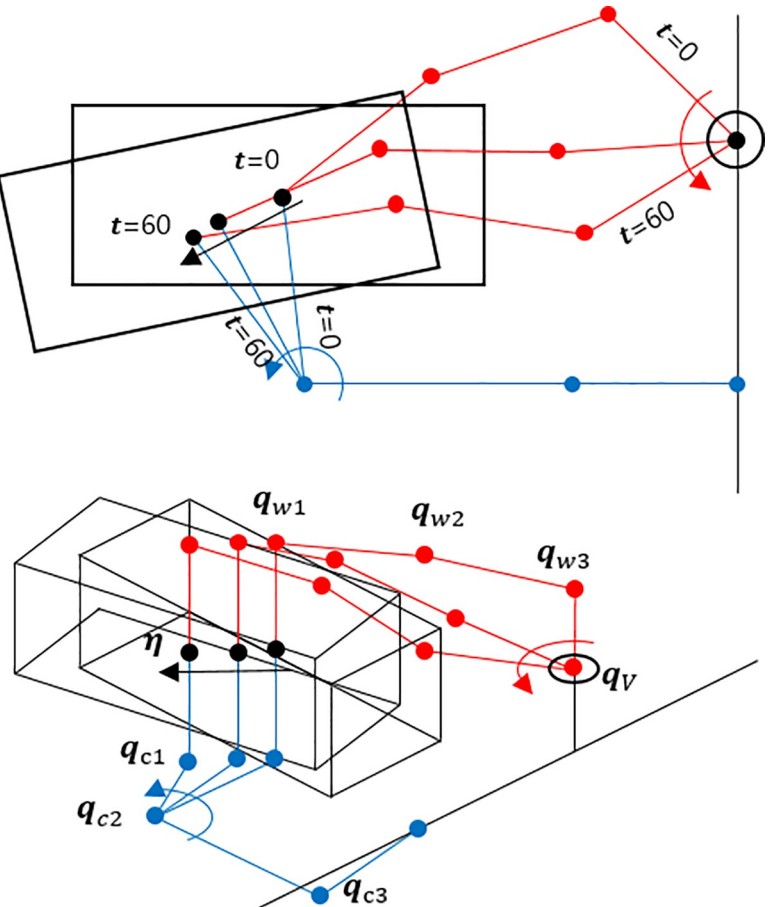

**Fig 4. The trajectory of the UVMS.**

We now use this constraint Jacobian to derive the forward kinematics of the UVMS. The position of AURORA is derived from the joint angle and link length of the working manipulator and clamping manipulator. The same position is derived for both working manipulator and clamping manipulator. Based on this, the constraint equation is expressed as

$$g(q_{all}) = [(\boldsymbol{\eta}_w - \boldsymbol{\eta})^{\mathrm{T}}, (\boldsymbol{\eta}_c^* - [x, y, z, \psi]^{\mathrm{T}})^{\mathrm{T}}]^{\mathrm{T}}. \tag{3}$$

$\boldsymbol{\eta}_w$ is the AUV's position and orientation vector obtained from the working manipulator $(q_w, l_{wi})$. $\boldsymbol{\eta}_c^*$ means the AUV's position and the yaw angle obtained from the clamping manipulator $(q_c, l_{ci})$. Because the working manipulator gripper makes surface-to-surface contact to maintain a horizontal plane when holding the valve, UVMS has a constraint that keeps it horizontal in an underwater situation. Therefore, the z-direction movement, roll, and pitch of this UVMS have no meaning, so they are omitted in this study. Other detailed kinematic studies have been dealt with in previous studies [14, 15], therefore, not covered in this study.

The dynamic equation [18] of a typical underwater robot body is derived as

$$\mathbf{M}_v \dot{\boldsymbol{v}} + \mathbf{C}_v(\boldsymbol{v})\boldsymbol{v} + \mathbf{D}_v(\boldsymbol{v})\boldsymbol{v} + \boldsymbol{g}_v = \boldsymbol{\tau}_v. \tag{4}$$

The detailed calculation results are presented in previous studies [40]. Based on a fixed ground coordinate system, the dynamic equation of the manipulator is derived as

$$\mathbf{M}_m\ddot{\boldsymbol{q}} + \mathbf{C}_m(\boldsymbol{q})\dot{\boldsymbol{q}} + \mathbf{D}_m(\boldsymbol{q})\dot{\boldsymbol{q}} + \boldsymbol{g}_m = \boldsymbol{\tau}_m. \tag{5}$$

$v$, $m$ are used as a subscript means AUV and manipulator. $\mathbf{M}_v$, $\mathbf{M}_m$ are the inertia matrices with added mass terms. $\mathbf{C}_v(\boldsymbol{v})$, $\mathbf{C}_m(\boldsymbol{q})$ are the centrifugal and Coriolis force. $\mathbf{D}_v(\boldsymbol{v})$, $\mathbf{D}_m(\boldsymbol{q})$ are the hydrodynamic damping matrices. $\boldsymbol{g}_v$, $\boldsymbol{g}_m$ are the gravity and buoyancy vector.

Because the AUV and the manipulator work together, we have to consider the interaction effects. Schjølberg combined each of the two models in his work by the Newton–Euler method. [18] The dynamics and interactions of UVMS were explained based on this combined model as

$$\mathbf{M}\dot{\boldsymbol{\zeta}} + \mathbf{C}\boldsymbol{\zeta} + \mathbf{D}\boldsymbol{\zeta} + \boldsymbol{g} = \boldsymbol{\tau}. \tag{6}$$

$\zeta$ is the body-fixed velocity of the UVMS. $\mathbf{M}$ and $\mathbf{D}$ are the mass inertia and the hydrodynamic drag force of the system; $\mathbf{C}$ is the centrifugal and Coriolis force of the UVMS; $\boldsymbol{g}$ is the gravity and buoyancy. In this study, centrifugal and Coriolis force matrices are also neglected to reduce the computational burden for real-time operation. The centrifugal and Coriolis forces have little effect if the motor speed is slow [41]. In order to reduce the computational burden, it is a general method of simplifying dynamics and ignores centrifugal and Coriolis forces in low-speed environments [42, 43]. Since UVMS was designed and adjusted in actual underwater conditions with neutral buoyancy, the gravity and buoyancy vector is ignored [40]. As a result, the dynamics of UVMS are summarized as

$$\mathbf{M}(\zeta)\dot{\boldsymbol{\zeta}} + \mathbf{D}(\boldsymbol{q},\dot{\boldsymbol{q}},\zeta)\boldsymbol{\zeta} = \boldsymbol{\tau}. \tag{7}$$

$$\mathbf{M}(\zeta) = \begin{bmatrix} \mathbf{M}_v + \mathbf{H}_w(\boldsymbol{q}_w) + \mathbf{H}_c(\boldsymbol{q}_c) & \mathbf{M}_{Cw}(\boldsymbol{q}_w) & \mathbf{M}_{Cc}(\boldsymbol{q}_c) \\ \mathbf{M}_{Cw}{}^T(\boldsymbol{q}_w) & \mathbf{M}_w(\boldsymbol{q}_w) & \mathbf{O} \\ \mathbf{M}_{Cc}{}^T(\boldsymbol{q}_c) & \mathbf{O} & \mathbf{M}_c(\boldsymbol{q}_c) \end{bmatrix} \tag{8}$$

$$\mathbf{D}(\boldsymbol{q},\zeta) = \begin{bmatrix} \mathbf{D}_v(\boldsymbol{v}) + \mathbf{D}_{w1} + \mathbf{D}_{c1} & \mathbf{D}_{w2} & \mathbf{D}_{c2} \\ \mathbf{D}_{w3} & \mathbf{D}_w & \mathbf{O} \\ \mathbf{D}_{c3} & \mathbf{O} & \mathbf{D}_c \end{bmatrix} \tag{9}$$

$$\boldsymbol{\zeta} = [\boldsymbol{v}^T, \boldsymbol{q}_w^T, \boldsymbol{q}_c^T]^T. \tag{10}$$

$\mathbf{M}_{Cw}$, $\mathbf{M}_{Cc}$ are the reaction force and the moment between the AUV and the working and clamping manipulators; $\mathbf{M}_w$, $\mathbf{M}_c$ are the mass inertia of the working and the clamping manipulators; $\mathbf{H}_w$, $\mathbf{H}_c$ are the inertia caused by attaching the working and the clamping manipulators; $\mathbf{D}_v$, $\mathbf{D}_{wi}$, $\mathbf{D}_{ci}$ are the drag terms resulting from the interaction between the AUV, the working, and the clamping manipulators, respectively.

The calculation results for this were introduced in previous studies [44]. The dynamic equation of the UVMS fixed to the body can be summarized as follows [45]:

$$\hat{\mathbf{M}}\ddot{\boldsymbol{q}}_u + \hat{\mathbf{D}}\dot{\boldsymbol{q}}_u = \boldsymbol{\Gamma}^T\boldsymbol{\tau}_r$$

$$(\hat{\mathbf{M}} = \boldsymbol{\Lambda}^T\mathbf{M}\boldsymbol{\Lambda}, \ \hat{\mathbf{D}} = \boldsymbol{\Lambda}^T\mathbf{M}\dot{\boldsymbol{\Lambda}} + \boldsymbol{\Lambda}^T\mathbf{D}\boldsymbol{\Lambda}). \tag{11}$$

$\tau_r$ is the force and torque matrix of the actuated joints and consist of $\tau_v$, $\tau_w$, and $\tau_c$. This means $\tau_r$ consists of the torque of AUV and the working and clamping manipulators. If we transform these equations to calculate $\ddot{\boldsymbol{q}}_u$, we obtain

$$\ddot{\boldsymbol{q}}_u = \hat{\mathbf{M}}^{-1}(\boldsymbol{\Gamma}^T\boldsymbol{\tau}_r - \hat{\mathbf{D}}\dot{\boldsymbol{q}}_u). \tag{12}$$

That is, it is possible to calculate the amount of angular acceleration at which the drive motor should move to achieve the target torque using this equation. This recalculates the constrained Jacobian to recalculate the angular behavior of the entire joint from the driving joint. Integrating the $\dot{\boldsymbol{q}}_{all}$ produced at a unit time allows the angle of all joints to be obtained; the manipulator is controlled using the determined angle value.

## 2.2. Desired trajectory and torque distribution

The control process of the manipulator is summarized in Fig 5. The subscript *desired* represents the theoretical target value. These desired values are based on previous studies comparing the method of turning the valve based on the joint torque [46]. In the previous study, three alternatives were proposed, and among them, the method of holding the valve and pipe was pointed out as the best method. In addition, angles of the valve and joint were calculated as optimal values with a trajectory that minimizes the velocity norm in the previous study. The detailed discussion on this is described in Bae's dissertation. In this study, experiments are conducted based on the best method identified. The manipulator control process consists of manipulator trajectory generation, proportional differential (PD) control, dynamics analysis, driving angle, and total angle conversion using the constrained Jacobian.

First, we set $\dot{\boldsymbol{q}}_{V.desired}$ to the speed at which the valve will move. In this study, valve operation was performed with a motion starting at 45˚ and rotating through 90˚. The valve motion was set to the valve rotation at 1/48 $\pi$ angular velocity in the counterclockwise direction, as shown in Fig 6. These values are set the same value in the previous experiment in order to compare with the previous studies according to the valve work of the fixed weight matrix [14].

To determine the optimal path to rotate the valve, the optimal criterion was to minimize the norm term of the speed of the actuated joints, as described in

$$\min\left(\|\dot{\boldsymbol{q}}_r\|\right). \tag{13}$$

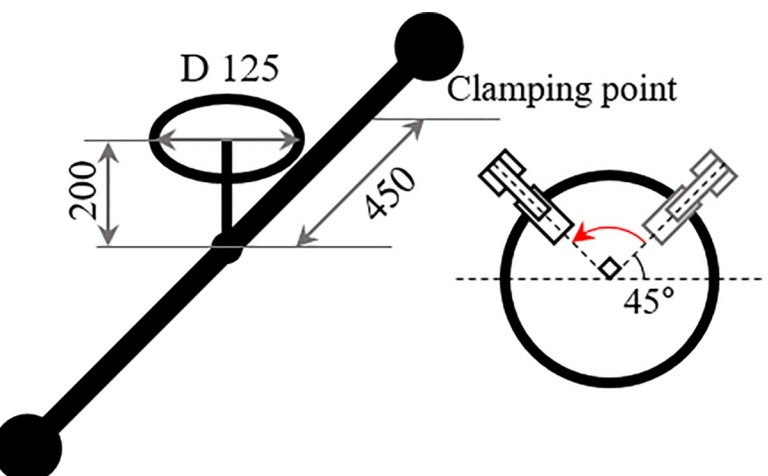

**Fig 5. Manipulator control flowchart.**

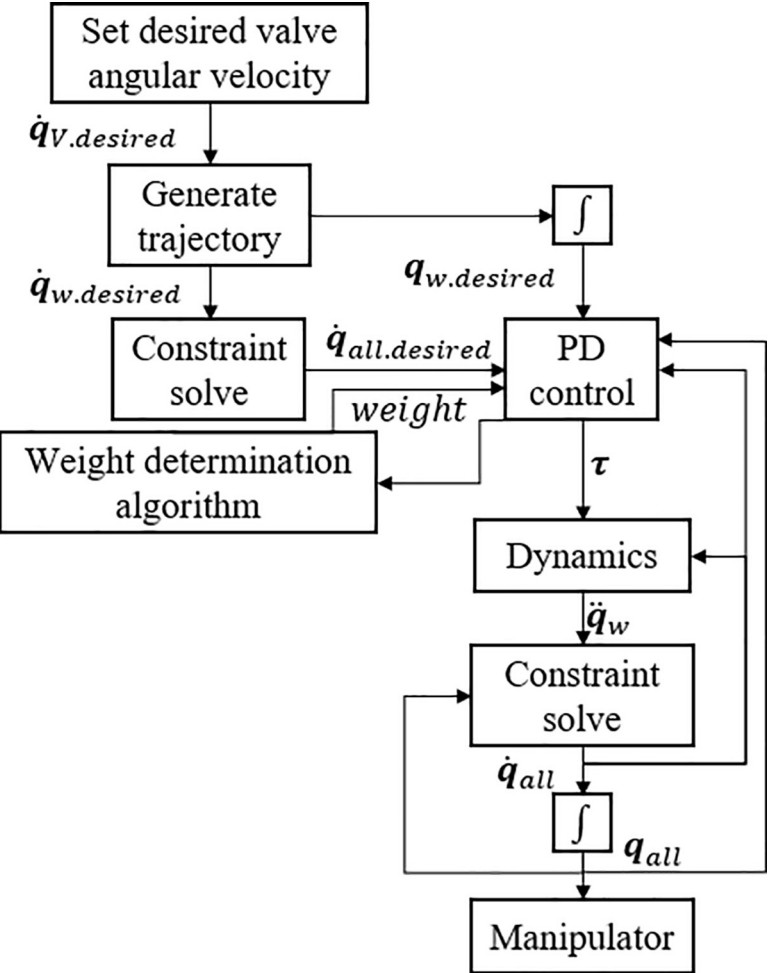

**Fig 6. Valve dimensions and rotation angle.**

Considering the definition of constraint Jacobian, the square of the norm term changes as follows:

$$\dot{\boldsymbol{q}}_r = \boldsymbol{\Gamma}\dot{\boldsymbol{q}}_u \tag{14}$$

$$\|\dot{\boldsymbol{q}}_r\|^2 = \dot{\boldsymbol{q}}_u^T\boldsymbol{\Gamma}^T\boldsymbol{\Gamma}\dot{\boldsymbol{q}}_u. \tag{15}$$

In addition, Eq (16) is established by the basic definition of Jacobian as

$$\dot{\boldsymbol{q}}_{u.desired} = \mathbf{J}_f^{\ddagger}\dot{\boldsymbol{q}}_{V.desired} \tag{16}$$

$$\mathbf{J}_f^{\ddagger} = (\boldsymbol{\Gamma}^T\boldsymbol{\Gamma})^{-1}(\mathbf{J}_f(\boldsymbol{\Gamma}^T\boldsymbol{\Gamma})^{-1})^{\dagger}. \tag{17}$$

$\dot{\boldsymbol{q}}_{u.desired}$, $\dot{\boldsymbol{q}}_{V.desired}$ are the desired angular velocities of the independent joints and the valve; $\mathbf{J}_f$, $\mathbf{J}_f^{\ddagger}$ are the forward Jacobian and the weighted pseudoinverse of the manipulators, respectively. That is, when $\dot{\boldsymbol{q}}_{V.desired}$ is determined, $\dot{\boldsymbol{q}}_{u.desired}$ is also determined.

When $\dot{\boldsymbol{q}}_{u.desired}$ is determined, the trajectory of the manipulator is determined. Based on this trajectory, torque distribution and manipulator operation are performed. Once $\dot{\boldsymbol{q}}_{u.desired}$ has

been determined, the restrained Jacobian can be used to determine all the angular velocities denoted by $\dot{\boldsymbol{q}}_{all}$ as [44]

$$\dot{\boldsymbol{q}}_{all} = \boldsymbol{\Lambda}\dot{\boldsymbol{q}}_u. \tag{18}$$

PD control is possible because $\dot{\boldsymbol{e}} = \dot{\boldsymbol{q}}_{u.desired} - \dot{\boldsymbol{q}}_u$ can be calculated when $\dot{\boldsymbol{q}}_{all}$ is derived and $\boldsymbol{e} = \boldsymbol{q}_{u.desired} - \boldsymbol{q}_u$ can be calculated by integrating per unit time. Cheng researched and applied the PD control model of redundantly actuated parallel manipulators [45] as

$$\boldsymbol{\tau}_{r.desired} = (\boldsymbol{\Gamma}^T)^{\#}(\mathbf{K}_D\dot{\boldsymbol{e}} + \mathbf{K}_P\boldsymbol{e})$$

$$(\boldsymbol{\Gamma}^T)^{\#} = \mathbf{W}^{-1}(\boldsymbol{\Gamma}^T\mathbf{W}^{-1})^{\dagger}. \tag{19}$$

\# indicates a weighted pseudoinverse. $\boldsymbol{e}$, $\dot{\boldsymbol{e}}$ are the errors in the independent joint angle and the angular velocity. Matrices $\mathbf{K}_D$ and $\mathbf{K}_P$ denote the differential and proportional gains in PD control, respectively. If $\boldsymbol{\tau}_{r.desired}$ is derived, it is possible to calculate the angular acceleration $\ddot{\boldsymbol{q}}_u$ of the driving joint that inverses the manipulator dynamics equation to generate a torque corresponding to $\boldsymbol{\tau}_{r.desired}$. Since this UVMS is the same as the UVMS used in the previous study, PD control was also used as the same [15].

$\mathbf{W}$, the weighting matrix, is the matrix for cooperation in the actuated joints as

$$\mathbf{W} = \begin{bmatrix} \mathbf{W}_v & \mathbf{O} & \mathbf{O} \\ \mathbf{O} & \mathbf{W}_w & \mathbf{O} \\ \mathbf{O} & \mathbf{O} & \mathbf{W}_c \end{bmatrix} \tag{20}$$

$$\mathbf{W}_v = \text{diag}(w_x, w_y, w_z, w_\phi, w_\theta, w_\psi) \tag{21}$$

$$\mathbf{W}_w = \text{diag}(w_{w1}, w_{w2}, w_{w3}) \tag{22}$$

$$\mathbf{W}_c = \text{diag}(w_{c1}, w_{c2}, w_{c3}). \tag{23}$$

$\mathbf{W}_v$, $\mathbf{W}_w$, and $\mathbf{W}_c$ are the weighting matrices of the AUV, working manipulator, and clamping manipulator; $w_x$, $w_y$, $w_z$ are the weighting matrix elements about the $x$, $y$, $z$ forces; $w_\phi$, $w_\theta$, $w_\psi$ are the weighting matrix elements about $\phi$, $\theta$, $\psi$ torques, respectively.

$w_{wi}$, $w_{ci}$ are the weighting matrix elements of the $i$-th joint of working and clamping manipulators, respectively. By adjusting the elements of the weighting matrix, the operator can control the force and torque of each actuated component. As the element of the weighting matrix increases, the torque or force of the component decreases.

## 3. The real-time torque distribution algorithm

### 3.1. Taguchi approach and problem formulation

DOE refers to a technique that defines and investigates all the possible conditions in an experiment involving multiple factors. To assist in the application of DOE, Taguchi proposed a series of techniques. [34] The first technique is the system design and the parameter design. It is also called problem formulation. It is based on the engineer's judgment of the parameters based on the current technology. System design helps identify the level of design factor, while parametric design helps determine the factor level related to the best performance. The factors that have a great influence on performance and can change during an experiment are called design variables. To change this design variable and observe the result, it is necessary to determine the

level of the design variable. The result value that is intended to approach the maximum, minimum, or a specific value through the process is called the objective function.

In addition, Taguchi presented the standardized DOE. For the experimental design, Taguchi used a special set of tables called orthogonal arrays (OAs), which represent the smallest partial factors and are used in common experimental designs. OA allows the selection of the necessary experiment using verticality from the entire experiment. OA is selected according to the degrees of freedom of the design variable level. Depending on the selected OA, the procedure to change the design variables was decided. After the standardized DOE, the experiment was conducted in the designated user condition based on the OA configured during the experiment plan.

The use of S/N analysis is proposed for better analysis of results. Using the S/N ratio to analyze the DOE results is easier while analyzing the results of a multi-sample test. The conversion of the S/N ratio depends on the objective function. It needs to be determined whether it should be small, large, or a specific value, as shown in Table 2. The S/N ratio is an indicator of performance, and a higher S/N ratio indicates better performance. It can be said that the level of the design variable in which the S/N ratio is high is a better design variable. In addition, the logarithmic transformation of the results of S/N ratios enables the prediction of performance improvement.

In this study, we present an algorithm that can increase the cooperation performance of the manipulator and the AURORA using the DOE and Taguchi methods. In a previous study, a method of dividing the burden on the manipulator's joint with an AUV was proposed. The burden on the manipulator can be reduced by the weighting matrix. The weighting matrix is the ratio of the burden between the driving elements. The higher the number, the lower the burden on the corresponding driving elements. Increasing the performance of the method means reducing the manipulator torque by increasing the force of the AUV thruster. The ratio at which this torque can be minimized changes every time depending on the position of the robot and the underwater scenario. It is difficult to check the position and condition of a robot because it is difficult to determine the weighting matrix in an underwater environment where the sensors and measuring devices are limited. The purpose of this study is to propose a method to determine the optimal distribution ratio between the driving elements in every situation where it is difficult to determine the cause of the errors.

As the first step in the Taguchi method, to proceed with problem formulation, the design variables and the objective function are set. The design variables are 12 positive elements of the weighting matrix. It consists of $\mathbf{W}_v$, $\mathbf{W}_w$ and $\mathbf{W}_c$. To avoid concentrating force on some joints, the sum of squares of torques is used as an objective function $y$. Because this objective function is in the form of variance, the overall torque is reduced. The problem formulation is

**Table 2. S/N ratio according to the objective function value.**

| Smaller-the-better (STB), | $SN_{STB} = -10 \, log_{10}(\frac{1}{n}\sum_{I=1}^{n} y_i^2)$ |
|---|---|
| Nominal-the-best (NTB), | $SN_{NTB} = -10 \, log_{10}\left(\frac{1}{n-1}\left(\sum_{I=1}^{n}(e_i^2) - S\right)\right),$ |
| | $S = \frac{1}{n}\left(\sum_{I=1}^{n}(e_i)\right)^2$ |
| Larger-the-better (LTB) | $SN_{LTB} = -10 \, log_{10}(\frac{1}{n}\sum_{I=1}^{n}\frac{1}{y_i}^2)$ |

summarized as follows:

$$\text{Given } \mathbf{W} = \begin{bmatrix} \mathbf{W}_v & \mathbf{O} & \mathbf{O} \\ \mathbf{O} & \mathbf{W}_w & \mathbf{O} \\ \mathbf{O} & \mathbf{O} & \mathbf{W}_c \end{bmatrix} \tag{24}$$

$$\text{Minimize } y = \sum_{i=1}^{3}(\tau_{ci}^2 + \tau_{wi}^2) \tag{25}$$

$$\text{Subject to } \mathbf{W} > 0$$

$$\boldsymbol{\tau}_{r.desired} = (\boldsymbol{\Gamma}^T)^{\#}(\mathbf{K}_D\dot{\boldsymbol{e}} + \mathbf{K}_P\boldsymbol{e}) \tag{26}$$

$$(\boldsymbol{\Gamma}^T)^{\#} = \mathbf{W}^{-1}(\boldsymbol{\Gamma}^T\mathbf{W}^{-1})^{\dagger}.$$

## 3.2. Real-time torque squared sum minimization algorithm

This algorithm was constructed based on the Taguchi method to lower the burden on the manipulator. The entire process of the weighting matrix determination algorithm is shown in Fig 7. The first step of the algorithm is to change the design variable, that is, the weighting matrix, from OA to the specified level. After calculating the torque and the force using the

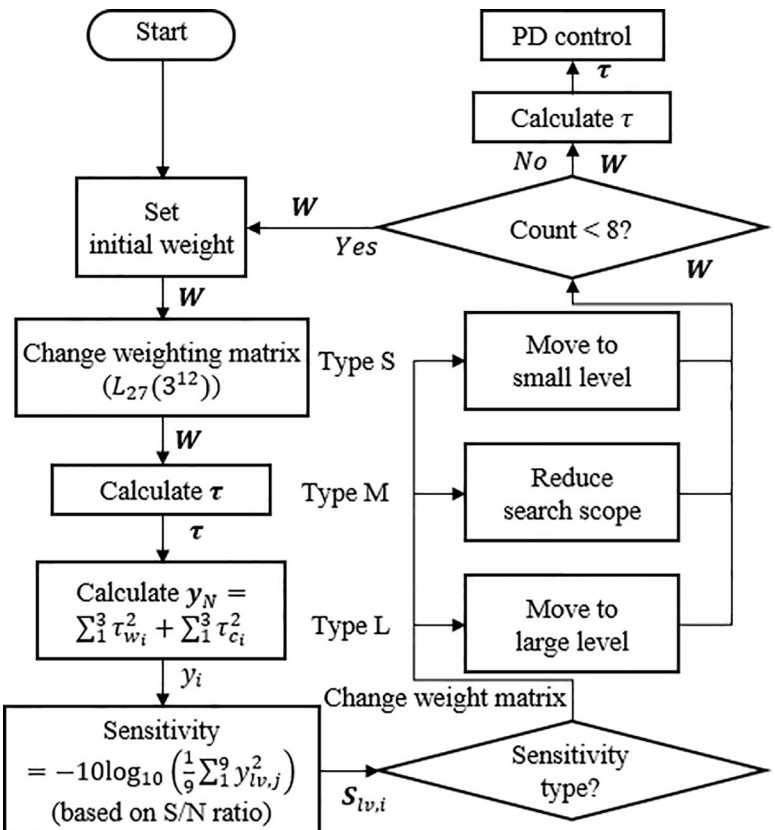

**Fig 7. Weighting matrix determination flowchart.**

changed weighting matrix, the sum of squares of the torque, and the objective function are calculated. Sensitivity, which was calculated using this objective function, is a novel definition and was determined based on the formula of the S/N ratio.

After the sensitivity analysis for each level of the weighting matrix, the level of the weighting matrix with high sensitivity is set as an element of the weighting matrix in the next experiment. Because it is based on the Taguchi method, the torque distribution performance increases as this process are repeated. However, the number of iterations should be limited so that there is no delay because the weighting matrix has to be calculated every second. Considering the performance of the actual device, in this study, the number of iterations of the experiment was determined eight times. In this algorithm, an experiment is a calculation in a program, not an actual physical experiment. As the process corresponding to this calculation is called an experiment in the Taguchi method, it has been called an experiment in this study also.

The design variable of this algorithm is the weighting matrix. It is composed of the weight of the AUV's degrees of freedom and the weight of the joint torque of the manipulator. It is called $w_i$ for convenience and $w_i$ is composed of 12 $w_1 \sim w_{12}$, as

$$\mathbf{W} = \mathrm{diag}[w_1, w_2, w_3, w_4, w_5, w_6, w_7, w_8, w_9, w_{10}, w_{11}, w_{12}]$$

$$= \mathrm{diag}[w_x, w_y, w_z, w_\phi, w_\theta, w_\psi, w_{W1}, w_{W2}, w_{W3}, w_{C1}, w_{C2}, w_{C3}]. \tag{27}$$

To determine the design variable, which can increase the performance, each design variable consisted of three levels. These levels were referred to as the increasing level, origin level, and decreasing level. If the level is larger than the 3 levels, it is possible to set a more precise range. However, since the calculation time is limited, the number of iterations decreases as the number of calculations increases. Therefore, instead of reducing the level, we set it to 3 levels to repeat more. Because there are 12 weighting matrix elements and each element has three levels, OA is determined using $L_{27}(3^{12})$ [47, 48]. The determined OA is shown in Table 3.

The torque and force are calculated because the weighting matrix is changed by OA. The weighting matrix is changed to three levels based on the previous weighting matrix. Through OA, 27 weighting matrices were constructed. The numbers up to 27 generated based on OA are called the experimental number N. These 27 experiments can suffice for all the experiments required for the sensitivity analysis to be conducted later. In the first run of the algorithm, the previous weighting matrix starts with the matrix used in the previous study. The fixed weighting matrix used in previous work is diag(141; 141; 100; 614; 265; 187; 1140; 1140; 1140; 1140; 1140; 1140;). The change in the weighting matrix according to each decreasing, original, and increasing level is shown as

$$w_{N,i} = \begin{cases} w_{prv-i} \times (1 - d_i) & \text{if } OA_{N,i} \text{ is } D \\ w_{prv-i} \times (1 - d_i) & \text{if } OA_{N,i} \text{ is } O \,. \\ w_{prv-i} \times (1 + d_i) & \text{if } OA_{N,i} \text{ is } I \end{cases} \tag{28}$$

$w_{prv-i}$ is an element of the previous weighting matrix $\mathbf{W}_{prv}$ and $w_{N,i}$ is an element of the newly changed weighting matrix $\mathbf{W}_N$. $\mathbf{W}_{prv}$ and $\mathbf{W}_N$ are the diagnostic matrices; $d_i$ is the ratio of the search range width. At the decreasing or increasing level, the previous weighting matrix $\mathbf{W}_{prv}$ is decreased or increased using a search range width $d_i$. The original level is set similar to the previous weighting matrix $\mathbf{W}_{prv}$. Through three levels, it is determined whether to increase or decrease the weighting matrix from the current value. For example, in the seventh experiment $w_4$, when the initial $w_{prv-4}$ was 614 and $d_i$ was 0.5, $w_{7,4}$ became 307, as the level was D

**Table 3. Orthogonal array for weight calculation.**

| N\i | 1 | 2 | 3 | 4 | 5 | 6 | 7 | 8 | 9 | 10 | 11 | 12 |
|---|---|---|---|---|---|---|---|---|---|---|---|---|
| 1 | D | D | D | D | D | D | D | D | D | D | D | D |
| 2 | D | D | D | O | O | O | I | I | I | D | O | I |
| 3 | D | D | D | I | I | I | O | O | O | D | I | O |
| 4 | D | O | I | D | O | I | D | O | I | O | D | I |
| 5 | D | O | I | O | I | D | I | D | O | O | O | O |
| 6 | D | O | I | I | D | O | O | I | D | O | I | D |
| 7 | D | I | O | D | I | O | D | I | O | I | D | O |
| 8 | D | I | O | O | D | I | I | O | D | I | O | D |
| 9 | D | I | O | I | O | D | O | D | I | I | I | I |
| 10 | O | D | I | D | I | O | I | O | D | I | I | I |
| 11 | O | D | I | O | D | I | O | D | I | I | D | O |
| 12 | O | D | I | I | O | D | D | I | O | I | O | D |
| 13 | O | O | O | D | D | D | I | I | I | D | I | O |
| 14 | O | O | O | O | O | O | O | O | O | D | D | D |
| 15 | O | O | O | I | I | I | D | D | D | D | O | I |
| 16 | O | I | D | D | O | I | I | D | O | O | I | D |
| 17 | O | I | D | O | I | D | O | I | D | O | D | I |
| 18 | O | I | D | I | D | O | D | O | I | O | O | O |
| 19 | I | D | O | D | O | I | O | I | D | O | O | O |
| 20 | I | D | O | O | I | D | D | O | I | O | I | D |
| 21 | I | D | O | I | D | O | I | D | O | O | D | I |
| 22 | I | O | D | D | I | O | O | O | D | I | O | D |
| 23 | I | O | D | O | D | I | D | I | O | I | I | I |
| 24 | I | O | D | I | O | D | I | O | D | I | D | O |
| 25 | I | I | I | D | D | D | O | O | O | D | O | I |
| 26 | I | I | I | O | O | O | D | D | D | D | I | O |
| 27 | I | I | I | I | I | I | I | I | I | D | D | D |

D: decreasing level, O: original level, I: increasing level.

[D,O,I] is same [−1,0,1] in algorithm.

(decreasing level). As another example, when the search range width was 0.5 with a fixed weighting matrix in the previous study, each level setting is summarized in Table 4.

After the 27 weighting matrices were generated, the torque of each joint was calculated using the determined weighting matrices. This process is defined by the *CalTorque* function in the algorithm. The joint torque calculation using Eq (19) summarized in Section 2, which is similar as

$$(\mathbf{\Gamma}^T)^{\#} = \mathbf{W}_N^{-1}(\mathbf{\Gamma}^T\mathbf{W}_N^{-1})^{\dagger}$$

$$\boldsymbol{\tau}_N = (\mathbf{\Gamma}^T)^{\#}(\mathbf{K}_D\dot{\boldsymbol{e}}_r + \mathbf{K}_P\boldsymbol{e}_r). \tag{29}$$

Because $\boldsymbol{e}_r$, $\dot{\boldsymbol{e}}_r$ and $\mathbf{\Gamma}$ are currently calculated and determined using the manipulator's kinematic parameter, the torque can be calculated when $\mathbf{W}_N$ is determined. Using the 27 weighted matrices, 27 torque matrices $\boldsymbol{\tau}_N$ were calculated. $\boldsymbol{e}_r$ is an error of the independent joints angle. $\boldsymbol{\tau}_N = [F_{N,x}, F_{N,y}, F_{N,z}, \tau_{N,\phi}, \tau_{N,\theta}, \tau_{N,\psi}, \tau_{N,w1}, \tau_{N,w2}, \tau_{N,w3}\ \tau_{N,c1}, \tau_{N,c2}, \tau_{N,c3}]$ is force and torque of UVMS for the next on $N$-th experiment and has the 12 torques and forces as elements.

**Table 4. Set initial value and initial stage of weighting matrix.**

| Design variable | | Level 1 [decreasing] | Level 2 [original] | Level 3 [increasing] |
|---|---|---|---|---|
| $w_1$ | $F_x$ | 70.5 | 141 | 211.5 |
| $w_2$ | $F_y$ | 70.5 | 141 | 211.5 |
| $w_3$ | $F_z$ | 50 | 100 | 150 |
| $w_4$ | $\tau_\emptyset$ | 307 | 614 | 921 |
| $w_5$ | $\tau_\theta$ | 132.5 | 265 | 397.5 |
| $w_6$ | $\tau_\varphi$ | 93.5 | 187 | 280.5 |
| $w_7$ | W1 | 570 | 1140 | 1710 |
| $w_8$ | W2 | 570 | 1140 | 1710 |
| $w_9$ | W3 | 570 | 1140 | 1710 |
| $w_{10}$ | C1 | 570 | 1140 | 1710 |
| $w_{11}$ | C2 | 570 | 1140 | 1710 |
| $w_{12}$ | C3 | 570 | 1140 | 1710 |

The next step is to calculate the objective function $y$ is composed of $y_1 \sim y_N$ which is defined to reduce the entire manipulator torque using the calculated torque. This process is defined by the *ObjFunction* function in the algorithm. The objective function is defined as the sum of the squares of each manipulator joint torque and is calculated using the sum of squares of torques of the working and clamping manipulators, as shown as

$$y_N = \sum_{i=1}^{3} \tau_{N,wi}^2 + \tau_{N,ci}^2. \tag{30}$$

Sensitivity analysis is used to evaluate the levels of design variables that can lead to maximum performance [33]. Level average analysis, generally used for sensitivity analysis, is a widely used method for optimal design. It calculates the average of the objective functions corresponding to the level of the design variable. It has been observed that the larger the average size, the higher the derived performance. We applied the S/N ratio from the Taguchi method to level the average analysis. In the Taguchi method, an index that can evaluate the performance of all the objective function cases is as follows: "bigger is better," "smaller is better," "nominal is best." It is called the S/N ratio and shown as

$$S/N \ ratio = -10 \log_{10}(\frac{1}{n}\sum_{i=1}^{n} y_i^2). \tag{31}$$

We have combined the S/N ratio from the "smaller is better" case with the level average analysis because the lower the total torque, the higher is the performance. The sensitivities $S_{lv,i}$ corresponding to the level of each design variable is calculated by applying the S/N ratio to the objective function $y$ using Eq (32)

$$\boldsymbol{S}_{lv,i} = -10 \log_{10} \frac{1}{9} \sum_{j=1}^{9} y_{lv,j}^2. \tag{32}$$

$\boldsymbol{S}_{lv,i}$ is the sensitivity of the *lv* level of design variable $w_i$. It is calculated using, $y_{lv,j}$ which means that 9 $y_N$ corresponds to the *lv* level of design variable $w_i$. Selecting the $y_{lv,j}$ process is defined by the *CategorizeLV* function in the algorithm. As there are three levels for 27 experiments, the number of $y_N$ is 9. Because it is calculated as a simple sum of squares, it is meaningless even if the order of j changes. $1/9 \sum_{j=1}^{9} y_{lv,j}^2$ is the mean square deviation (MSD), an index that reflects the mean and standard deviation of the data. MSD was defined according to the

S/N ratio of the Taguchi method. The logarithmic to base 10 transformations was applied to view a wide range of data and the linearity of the influence conveniently. Because the log is a monotonic function, it does not affect the results of the sensitivity analysis.

Based on this sensitivity analysis, the aforementioned process was repeated. After calculating the sensitivities according to the three levels of the 12 design variables, a design variable level that improved the performance with sensitivity was determined. If the sensitivity is maximum at the decreasing or increasing level, the performance increases as the weighting matrix decreases or increases. Therefore, the weighting matrix in the next process is changed to the weighting matrix set at the decreasing or increasing level. In the case of increasing or decreasing levels, the weighting matrices are changed to a higher or lower value. These cases are called type L and S for convenience. If the sensitivity is maximum at the original level, the search range $d_i$ reduces. This means that the optimal value of the design variable exists between the decreasing and increasing levels. The reduction ratio of $d_i$ is called the $S_{\text{scale}}$, and in this algorithm is set to 0.5. The case where the search range is reduced using this procedure is defined as type M. In type M, the weighting matrix does not change.

A new weighting matrix was determined through sensitivity analysis, and the previous process was repeated using it. The newly determined weighting matrix becomes the previous weighting matrix $\mathbf{W}_{prv}$ in the next iteration. In this way, a weighting matrix is determined that increases the performance through repetition. The series of processes is repeated eight times in this algorithm; the resulting weighting matrix exhibits the highest performance that can be calculated in real-time. The weighting matrix is determined once every 50 ms, which is a control period through the eight iterations. Because the valve rotation operation of this algorithm takes 60 s, the weighting matrix is determined 1200 times during the operation. The entire algorithm is described in Table 5.

## 4. Experiment and results analysis

### 4.1. Experiment setup

To show that the algorithm is effective in real situations, experiments were conducted using robots and test benches used in previous studies. The test bench pipe was attached using a suction device on the wall of the glass tank. The overall view of the test bench is shown in Fig 8. The valve is located at the center of the pipe. If the valve is in the water for a long period, it can rust. Rust increases the friction in the valve rotation operation, which is an error factor in the valve operation. The actual UVMS experiment is shown in Fig 9 and the position of the valve, clamping point, and diameter of the valve are depicted in Fig 6.

Measurements during valve operation are the angular error and torque value of each joint. It is important to ensure that the valve operation is in the desired trajectory. Therefore, the error is calculated by subtracting the desired joint angle from the current joint angle. An encoder is combined with the joint motor of the manipulator, and the encoder measures the current joint angle. The encoder is built into the Maxon EC 60 flat. The joint torque sensor measures the workload on the manipulator's joints during valve operation. The torque sensor is connected to the motor inside the joint and measures the degree of twist. The torque sensor is a special order product of SETEC, and the maximum measurable torque is about 20Nm. The internal structure of the joint is depicted in Fig 10. The length of each link of the UVMS used in the experiment is depicted in Table 6.

The process of the experiment was divided into robot submersion, assembly, setting, clamping, and execution of valve turning. First, the robots were placed in a water tank, and a manipulator and body were assembled in the water. After moving the assembled robot to the control position, the pose and parameters of the robot were initialized. The gripper's pneumatic switch

**Table 5. Real-time Taguchi weight pseudocode.**

$N_{Loop}$ = 8; // Number of loop for one determined weighting matrix
$N_w$ = 12; // Number of weight elements
$N_{exp}$ = 27; // Number of total experiment from OA
$N_{lev}$ = 9; // Number of experiment per each level
// $\mathbf{W}_{prev}$: Determined weighting matrix in the previous loop [12*12]
// $\mathbf{W}_{temp}$: Temporal changed weighting matrix by OA [12*12]
// **QA**: Orthogonal array [27*12]
// $S_{Max}$: Maximum of sensibility
// $\mathbf{W}_{Final}$: Final determined weighting matrix
// $\tau_{Final}$: Torque and force of UVMS based on $\mathbf{W}_{Final}$
**Input:** $\mathbf{W}_{prev}, \boldsymbol{e}_r, \dot{\boldsymbol{e}}_r$

| *for* | $n = 1$ to $N_{Loop}$ | |
|---|---|---|
| | ***for*** *i = 1 to $N_w$, N = 1 to $N_{exp}$, j = 1 to $N_{lev}$* | |
| | *// Change weighting matrix by OA* | |
| | $\mathbf{W}_{temp}(i,i) = \mathbf{W}_{prev}(i,i) (1 + \mathbf{OA} (N,i)^* d_i);$ | |
| | *// Calculate system torque of k-th case* | |
| | $\boldsymbol{\tau}_N = CalTorque(\mathbf{W}_{temp}, \boldsymbol{e}_r, \dot{\boldsymbol{e}}_r)$ | |
| | *// Calculate objective function* | |
| | $\boldsymbol{y} = ObjFunction(\boldsymbol{\tau}_N)$ | |
| | $y_{lv,j} = CategorizeLV(\boldsymbol{y})$ | |
| | *// Calculate sensibility of k-th case* | |
| | $S_{lv,i} = -10 \log_{10} \frac{1}{9} \sum_{j=1}^{9} y_{lv,j}^2;$ | |
| | *end* | |
| | *// Determine the maximum sensitivity* | |
| | $S_{Max} = Max(S_{lv,i})$ | |
| | *If* | level of $S_{Max}$ = = decreasing level *//S type (W go to decreasing level)* $\mathbf{W}_{prev}(i,i) = \mathbf{W}_{temp}(i,i)^* (1 - d_i);$ |
| | *elseif* | level of $S_{Max}$ = = original level *//M type (narrow search range)* $d_i = d_i^* S_{scale};$ |
| | *elseif* | level of $S_{Max}$ = = increasing level *//L type (W go to increasing level)* $\mathbf{W}_{prev}(i,i) = \mathbf{W}_{temp}(i,i)^* (1 +d_i);$ |
| | *end* | |
| *end* | | |

$\mathbf{W}_{Final} = \mathbf{W}_{prev};$
// Torque calculation with the determined weighting matrix
**Return** $\boldsymbol{\tau}_{Final} = CalTorque(\mathbf{W}_{Final}, \boldsymbol{e}_r, \dot{\boldsymbol{e}}_r)$

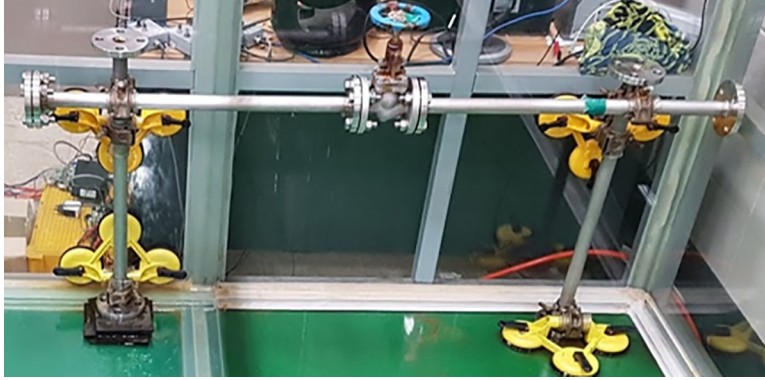

**Fig 8. Test bench and valve structures.**

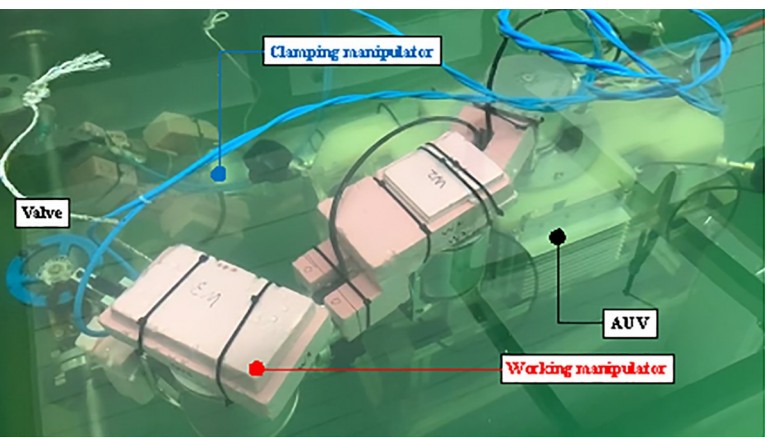

**Fig 9. Experimental setting in the test bench.**

was activated to clamp the valve of the manipulator. At this time, the torque sensor value was set as 0. Then, the valve was rotated 90° clockwise and the sensor values were recorded and analyzed to demonstrate the effectiveness of the algorithm.

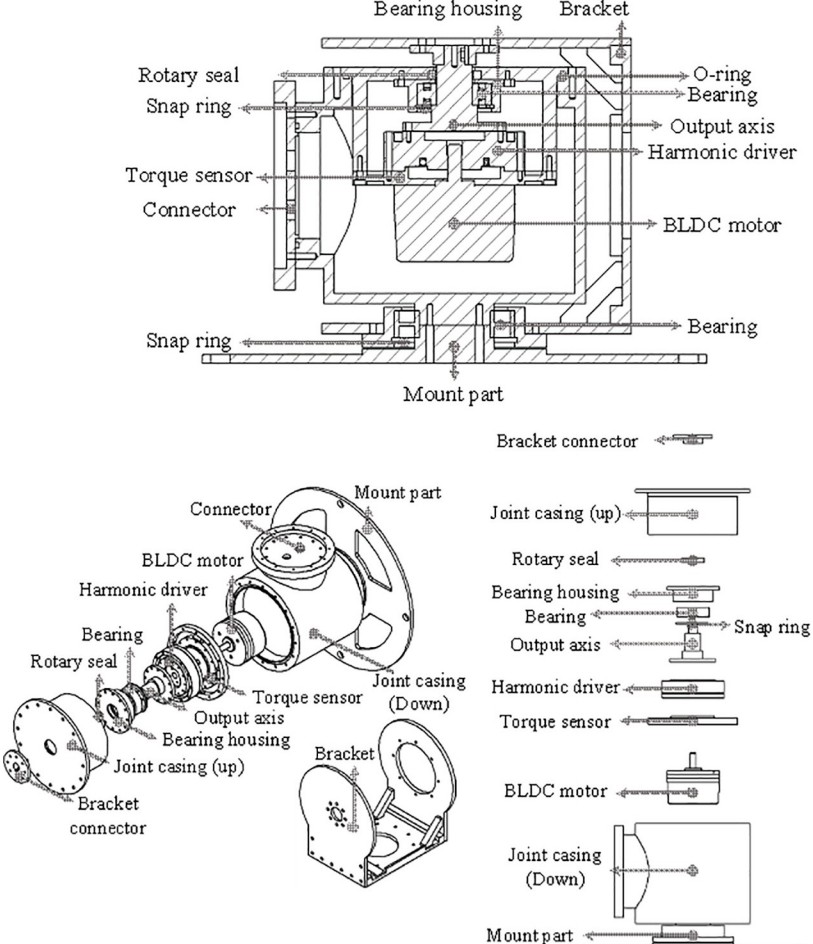

**Fig 10. Detailed description of a manipulator joint.**

**Table 6. Length of UVMS links.**

| $l_{w1}$ | $l_{w2}$ | $l_{w3}$ | $l_{c1}$ | $l_{c2}$ | $l_{c3}$ |
|---|---|---|---|---|---|
| 350 mm | 250 mm | 288 mm | 382 mm | 463 mm | 423 mm |

Two types of experiments were repeated five times each, and the experimental results were analyzed by calculating the average of torque sensor values. The first experiment was a valve turning about a fixed weighting matrix used in the previous study. Because the weighting matrix was fixed, the results of the iterative experiment appeared similarly. Therefore, we averaged the results of these iterative experiments to compare them with the changes in the algorithm.

The other experiments involved changing the real-time weighting matrix using the main algorithm. In the algorithm, the weighting matrix changed in addition to the torque, thrust, and weighting matrix of the joint, which changed every second. Therefore, we categorized the results into three cases: the best, the worst, and the average torque cases of the experiments. The best case and worst case represent the lowest and highest torque values, respectively. The average case is the average torque of each experiment while changing the weighting matrix experiment.

## 4.2. Experiment results of real-time weight matrix determination algorithm

The primary aim of the experiment was to rotate the valve by 90˚; the rotation angle error was compared to verify that the valve operation was performed correctly. Fig 11 shows the progress of the error in the valve angle during the valve operation for each case. The average case has a lower error during the operation compared to the fixed case. As shown in Table 7, the error in the valve angle in the worst case after valve operation is lower than that in the fixed case. In

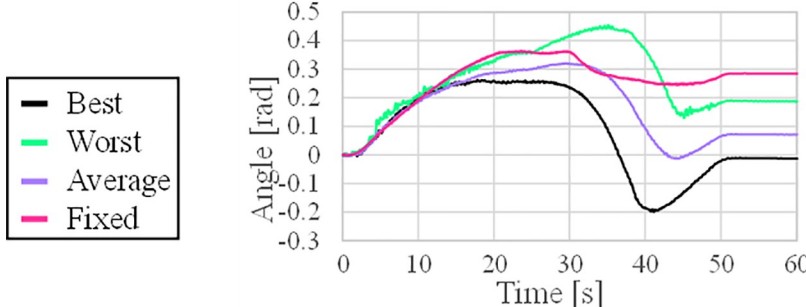

**Fig 11. Maximum value among manipulator joints over time.**

**Table 7. Decrease rate of valve angle error between fixed and average case.**

| Case | Max error | Error after valve turning |
|---|---|---|
| Fixed | 0.364 | 0.285 |
| Best | 0.262 | -0.011 |
| Worst | 0.453 | 0.188 |
| Average | 0.321 | 0.071 |

**Table 8. Max and final error in each case.**

| [%] | Max error | Error after valve turning |
|-----|-----------|---------------------------|
| Rate | 12% | 75% |

addition, as shown Table 8, the final error in the average case is reduced by approximately 75% compared to the fixed case. In summary, it can be seen that the accuracy of the valve operation is improved through the algorithm.

The torque applied to each joint is compared using a torque sensor built into the joint. The overall manipulator joint torque and the AUV thrust change overtime during the valve operation are shown in Fig 12. Fig 13 shows the maximum value of the total joint torque to understand the trend of the total torque in each case of the valve operation. In the average case, the maximum torque value of all the joints decreased compared to the fixed case. For a detailed analysis, the torque change in each joint was analyzed. The torque value of each joint is generally lower in the average case than in the fixed case.

For a more accurate comparison, we summarized the maximum torque values between the average and fixed cases. The maximum value of each joint torque during the valve operation is summarized as a bar chart in Fig 14, and it can be seen that all the values have decreased. The difference and the rate of change of the maximum torque and the force between the average and the fixed case are summarized in Tables 9 and 10. The torque of each manipulator has a very small value with a maximum of 1.4. It is because UVMS is in a neutral buoyancy state, so there is no force applied in the z-axis direction, and the force is used only to turn the valve in the xy-plane direction. Owing to the limit of the thruster, the maximum range of $F_x$ and $F_y$ is up to 145.6N. Therefore, the thrusters operate within the operating range.

According to the experimental results, it can be seen that the manipulator joint torque was reduced through the distribution of the thruster load. The torque of all the joints decreased from 38% to 55% as shown in Table 9. The maximum thrust force of the AUV increased by 21.3N, 109.7N, and 0.4Nm at $F_x$, $F_y$ and, $\tau_\psi$ respectively as show in Table 10. To summarize, this algorithm sufficiently lowered the manipulator torque by distributing the manipulator's burden with the AUV's thrust. Compared to the fixed weight case, the sway force was significantly increased. The bent manipulator stretches as the vehicle moves backward during valve operation. The vehicle turns while pulling the valve backward. The thruster generates this backward moving force, and this force replaces the manipulator's torque. As a result, instead of decreasing the torque, the sway force in the y-axis direction increases compared to the surge force in the x-axis direction.

## 4.3. Weighting matrix

To evaluate the effectiveness of this algorithm, we observed the weighting matrix every second during the experiment. The changes in each weighting matrix during the experiment are summarized in Fig 15. The average is the average value of the weighting matrix for four experiments. The maximum and minimum values of the weighting matrix for each case are summarized in Table 11. It can be seen that the range varies from 0.781–19477. Because the range is different from the weighting matrix of the fixed case, it is concluded that the value of the fixed case is not optimal. In addition, it can be seen that the weighting matrix changed every second. It is difficult to derive an optimal torque distribution with a fixed weighting matrix. Therefore, to derive the optimal torque distribution of the UVMS, this algorithm can be concluded to be effective because the weighting matrix must be changed in real-time in various ranges.

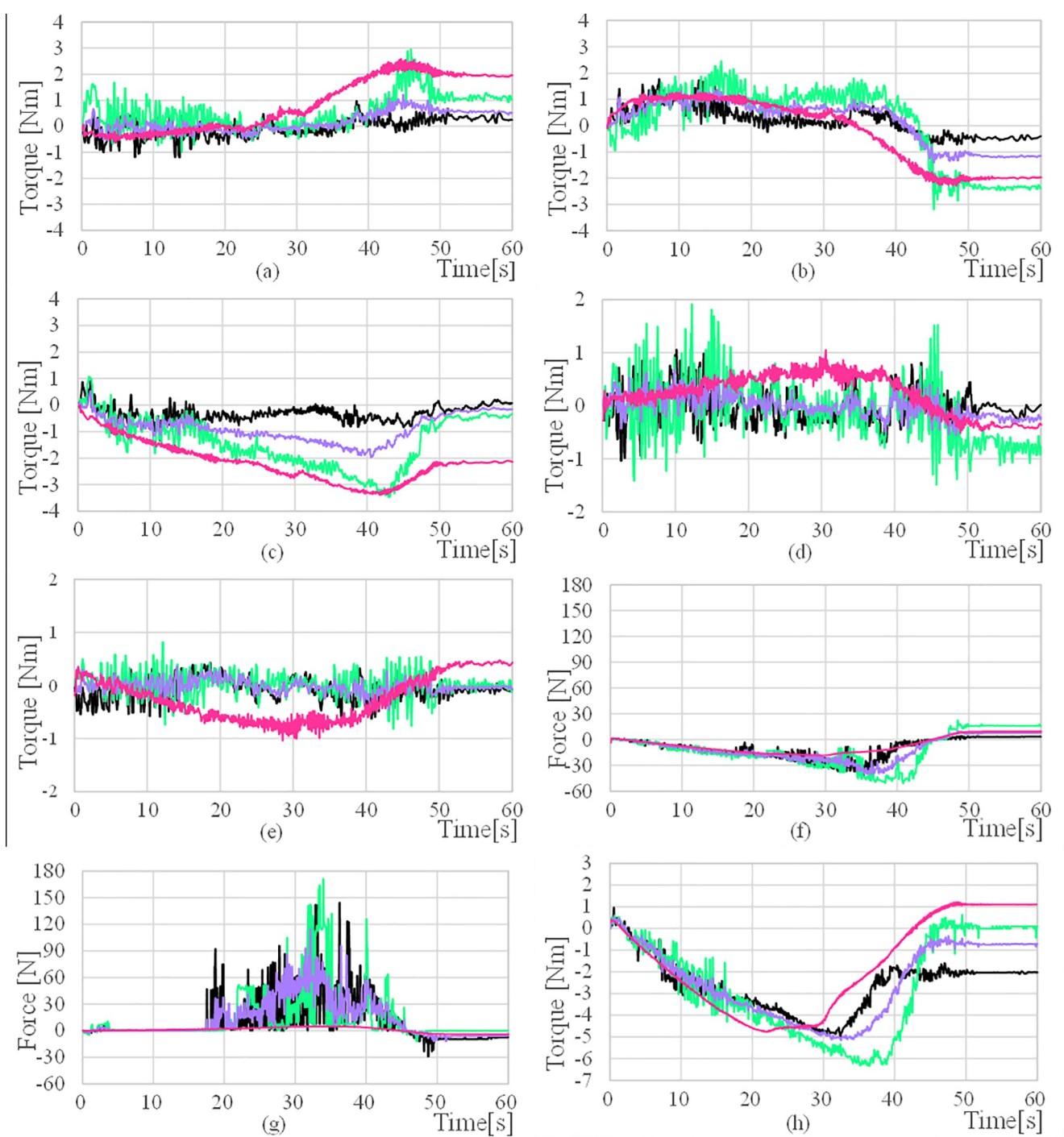

**Fig 12.** Manipulator joint torques and AURORA thruster force (a) $\tau_{W1}$ (b) $\tau_{W2}$, (c) $\tau_{W3}$ (d) $\tau_{C1}$ (e) $\tau_{C2}$ (f) $F_x$ (g) $F_y$ (h) $\tau_\psi$.

## 5. Conclusion

An algorithm for lowering the sum of squared torques of the manipulator joints using the Taguchi method to divide the manipulator torque with the thruster force of the AUV was presented and experimentally verified. The design variables and S/N ratio of the Taguchi method were used in real-time every second to minimize the torque sum of squares and reduce the

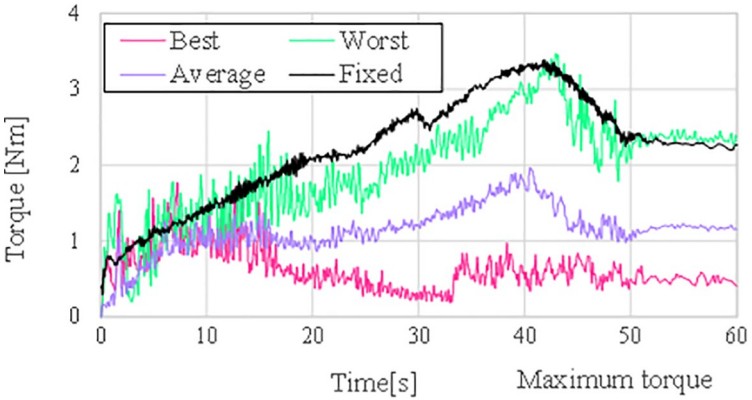

**Fig 13. Maximum value among manipulator joints over time.**

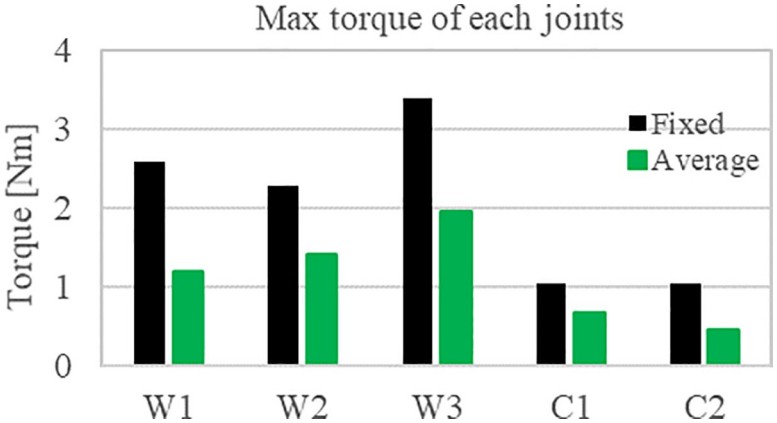

**Fig 14. Comparison of the maximum torque applied to the joint during valve operation.**

**Table 9. Average difference joint torque between fixed weight and algorithm cases.**

|  | W1 | W2 | W3 | C1 | C2 |
|---|---|---|---|---|---|
| Average [Nm] | -1.4 | - 0.9 | - 1.4 | - 0.4 | - 0.6 |
| Ratio [%] | - 53% | - 38% | - 42% | - 35% | - 55% |

**Table 10. Average difference thrust force and torque between fixed weight and algorithm cases.**

|  | $F_x$ | $F_y$ | $\tau_\psi$ |
|---|---|---|---|
| Average | + 21.3N | + 109.7N | + 0.4Nm |
| Ratio [%] | + 114.9% | + 2268.3% | + 7.9% |

calculation time. The optimal weighting matrix that can minimize the sum of squares of torque is repeatedly searched every second, and the control is performed to minimize the sum of squares of torque of the manipulator's joint.

To compare the fixed weighting matrix and the optimal weighting matrix selection algorithm, an experiment was conducted to measure the joint torque and thrust force of the AUV

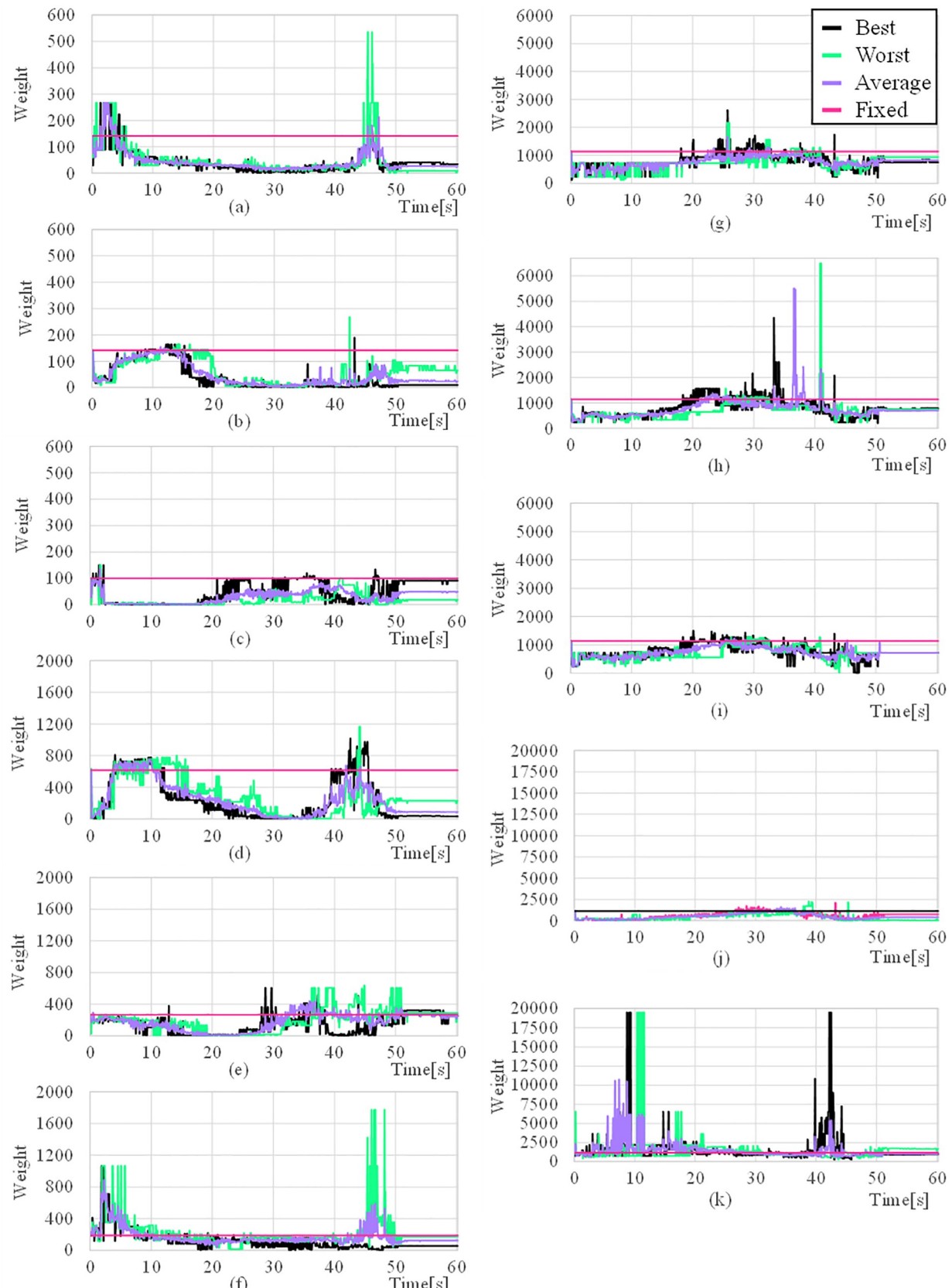

**Fig 15.** Weighting matrix elements (a) $w_x$ (b) $w_y$, (c) $w_z$ (d) $w_\phi$ (e) $w_\theta$ (f) $w_\psi$ (g) $w_{W1}$ (h) $w_{W2}$, (i) $w_{W3}$ (j) $w_{C1}$ (k) $w_{C2}$.

**Table 11. Max and min values of weighting matrix in each cases.**

|  | Best | Worst | Average | Fixed |
|---|---|---|---|---|
| Max | 19477 | 19477 | 12985 | 1140 |
| Min | 0.781 | 0.781 | 0.781 | 100 |

while rotating the underwater valve by 90˚. Through the experiment, the error of the angle that occurs during valve operation is reduced by an average of 75%. The thrust force of AUV increased by 114.9% in the x-axis direction and 2268.3% in the y-axis direction, and the torque of each joint decreased by 53%, 38%, 42% for the working manipulator and 35%, 55% for the clamping manipulator for each joint. In addition, the weight value of the Z-axis of the AUV, which has the smallest range of the optimal weight matrix found through the search, is from 0 to 200. The weight value of manipulator joint C2, which has the largest range of the optimal weight matrix found through the search, is from 0 to 20000. It means that there is a limit with the fixed weighting matrix to cooperative control between AUV and manipulator.

A future research plan is to control using a criterion other than the sum of squares of each torque. It is expected that better performance can be achieved by modifying the criteria and control method of the optimal weighting matrix such as fuzzy logic according to the absolute torque value.

## Author Contributions

**Conceptualization:** Yecheol Moon, Sangrok Jin, Jangho Bae, TaeWon Seo.

**Funding acquisition:** TaeWon Seo.

**Investigation:** Yecheol Moon, Jongin Hong.

**Project administration:** TaeWon Seo.

**Supervision:** Jangho Bae, TaeWon Seo.

**Writing – original draft:** Yecheol Moon, Jangho Bae, TaeWon Seo.

**Writing – review & editing:** Yecheol Moon.

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
