## [Decision Letter · Decision Letter 0]

19 Mar 2021

PONE-D-20-38676

Real-Time UVMS Torque Distribution Algorithm based on Weighting Matrix

PLOS ONE

Dear Dr. Seo,

Thank you for submitting your manuscript to PLOS ONE. After careful consideration, we feel that it has merit but does not fully meet PLOS ONE’s publication criteria as it currently stands. Therefore, we invite you to submit a revised version of the manuscript that addresses the points raised during the review process.

We look forward to receiving your revised manuscript.

Kind regards,

Mohammad Mehdi Rashidi

Academic Editor

PLOS ONE

Journal Requirements:

3. We note you have included a table to which you do not refer in the text of your manuscript. Please ensure that you refer to Table VI in your text; if accepted, production will need this reference to link the reader to the Table.

Reviewers' comments:

Reviewer's Responses to Questions

**Comments to the Author**

1. Is the manuscript technically sound, and do the data support the conclusions?

Reviewer #1: Yes

Reviewer #2: Yes

2. Has the statistical analysis been performed appropriately and rigorously? 

Reviewer #1: Yes

Reviewer #2: Yes

3. Have the authors made all data underlying the findings in their manuscript fully available?

Reviewer #1: Yes

Reviewer #2: Yes

4. Is the manuscript presented in an intelligible fashion and written in standard English?

Reviewer #1: Yes

Reviewer #2: No

5. Review Comments to the Author

Reviewer #1: The manuscript aimed at investigating a real-time algorithm for even distribution of the torque burden on the parallel manipulator with an autonomous underwater vehicle. The following comments should be addressed before possible publishing.

• Nomenclature should be added in order to define all variables and abbreviations used in the paper.

• Has the repeatability of the experiments been checked?

• The uncertainty of the experiments must be calculated.

• It is necessary to explain the characteristics of the test equipment in more detail.

Reviewer #2: This research presents a real-time algorithm for evenly distributing the torque burden on the parallel manipulator with an autonomous underwater vehicle (AUV) through the AUV and manipulator's cooperation.

This paper is useful as regarding its technical view. However, I have concerns regarding the manuscript. Here the authors can find my comments on this manuscript. The authors need to address the comments I mentioned in their work.

The English language requires improvements. Please read the text carefully and search for typos ("a" "," "the", Capital letter, space between word, etc). For example, I fixed some part of the abstract as following that the authors could find out what I mean:

" even distributing the torque "

"For the redundant resolution of the underwater vehicle "

"Analysis of the experimental results revealed that the manipulator torque load was greatly reduced due to the AUV load distribution. "

Please spell out acronyms the FIRST time you use them in the abstract AND in the body of the manuscript, which means from the Introduction onwards. Please spell out acronyms the FIRST time you use them in the body of the manuscript again, even if already spelled out in the abstract or title.

"We studied this cooperation of the underwater manipulator and the AUV" you should motivate why you say that, because it does not clearly follow out of the literature discussion that you have given.

figure 10 is not clear. Please increase the quality.

Why did you select such ranges of angles?

Quality of figures 12 ad 13 are low, and captions may be too small for the overall size of figures. Production should request higher quality figures. Color figures have been formatted so they are unclear when printed in black and white.

"The manipulator joint torque was reduced through the distribution of the thruster's load." This is strong and contentious statement without proof? (You have not made such a case based on literature) If this is your own conclusion, it is out of place here in same paragraph with statement. And should probably be part of the next paragraph that motives the reason for your research?

"The subscript desired represents the theoretical target value" What are the other feasible alternatives? What are the advantages of adopting this particular metric over others in this case? How will this affect the results? More details should be furnished.

Figure 10 is not clear. Please increase the quality.

All the basic equations, therefore, an appropriate reference is necessary. the values of some of the parameter are not stated.

The measures method in experimental study are in my opinion not clear. Please use a clear definition.

The conclusion section is too general, please add the main conclusion of the simulation studies.

Moreover, the manuscript could be substantially improved by relying and citing more on recent literatures about Experimental case studies and numerical modelling such as the followings:

https://doi.org/10.1016/j.oceaneng.2020.108455, , https://doi.org/10.1115/1.4049040, https://doi.org/10.1016/j.engfailanal.2020.104548, https://doi.org/10.1063/1.5113592, https://doi.org/10.1016/j.flowmeasinst.2020.101717

6. PLOS authors have the option to publish the peer review history of their article (what does this mean?). If published, this will include your full peer review and any attached files.

Reviewer #1: No

Reviewer #2: No

---

## [Author Response · Author response to Decision Letter 0]

18 Apr 2021

Thank you for the valuable comments on this article. We carefully revised the paper according to the comments. Please refer the attached revision summary and revised draft for details. The revised contents are highlighted.

---

## [Decision Letter · Decision Letter 1]

29 Apr 2021

PONE-D-20-38676R1

Real-Time UVMS Torque Distribution Algorithm based on Weighting Matrix

PLOS ONE

Dear Dr. Seo,

Thank you for submitting your manuscript to PLOS ONE. After careful consideration, we feel that it has merit but does not fully meet PLOS ONE’s publication criteria as it currently stands. Therefore, we invite you to submit a revised version of the manuscript that addresses the points raised during the review process.

We look forward to receiving your revised manuscript.

Kind regards,

Mohammad Mehdi Rashidi

Academic Editor

PLOS ONE

Journal Requirements:

Reviewers' comments:

Reviewer's Responses to Questions

**Comments to the Author**

1. If the authors have adequately addressed your comments raised in a previous round of review and you feel that this manuscript is now acceptable for publication, you may indicate that here to bypass the “Comments to the Author” section, enter your conflict of interest statement in the “Confidential to Editor” section, and submit your "Accept" recommendation.

Reviewer #1: All comments have been addressed

Reviewer #2: (No Response)

2. Is the manuscript technically sound, and do the data support the conclusions?

Reviewer #1: Yes

Reviewer #2: (No Response)

3. Has the statistical analysis been performed appropriately and rigorously? 

Reviewer #1: Yes

Reviewer #2: (No Response)

4. Have the authors made all data underlying the findings in their manuscript fully available?

Reviewer #1: Yes

Reviewer #2: (No Response)

5. Is the manuscript presented in an intelligible fashion and written in standard English?

Reviewer #1: Yes

Reviewer #2: (No Response)

6. Review Comments to the Author

Reviewer #1: All the comments are addressed.

Reviewer #2: The author did not answer all of my questions accordingly. They just have responded to some parts of issues and have left some others! I give one more chance to authors to answer ALL of my questions/ comments. If they cannot address all of my questions/ comments, I have no choice just to reject the paper.

7. PLOS authors have the option to publish the peer review history of their article (what does this mean?). If published, this will include your full peer review and any attached files.

Reviewer #1: No

Reviewer #2: No

---

## [Author Response · Author response to Decision Letter 1]

6 Jun 2021

Thank you for the valuable comments on this article. We carefully revised the paper according to the comments. The main corrections are as follows:

This review has responded to all comments of reviewer #2. All comments from Reviewer 1 were answered in the previous review. For the readability, only the comments of Reviewer 2 are summarized.

We checked and corrected grammatical errors in the paper.

The sentence that confuses the understanding of the paragraph has been corrected. 

To increase the quality of the figure, a description was added and inserted as a vector image. Also, the color has been changed for easy identification.

The contents of previous research necessary for understanding that were omitted in this paper were additionally described.

Previous research contents necessary for understanding has been added.

Added a description of the experimental settings.

The conclusion part has been rewritten to be more specific.

The reference recommended by Reviewer 2 was added to this review.

Please refer the attached revision summary and revised draft for details.

---

## [Editor Report · Decision Letter 2]

14 Jun 2021

Real-Time UVMS Torque Distribution Algorithm based on Weighting Matrix

PONE-D-20-38676R2

Dear Dr. Seo,

We’re pleased to inform you that your manuscript has been judged scientifically suitable for publication and will be formally accepted for publication once it meets all outstanding technical requirements.

Kind regards,

Mohammad Mehdi Rashidi

Academic Editor

PLOS ONE

Additional Editor Comments (optional):

The revised version could be accepted in the present form.

Associate Editor

M.M. Rashidi
---

## [Editor Report · Acceptance letter]

23 Jun 2021

PONE-D-20-38676R2 

Real-Time UVMS Torque Distribution Algorithm based on Weighting Matrix 

Dear Dr. Seo:

I'm pleased to inform you that your manuscript has been deemed suitable for publication in PLOS ONE. Congratulations! Your manuscript is now with our production department. 

Kind regards, 

on behalf of

Professor Mohammad Mehdi Rashidi 

Academic Editor

PLOS ONE